# The ribosome-inactivating proteins MAP30 and Momordin inhibit SARS-CoV-2

**Norman R. Watts**[1]⊕*, **Elif Eren**[1]⊕, **Ira Palmer**[1], **Paul L. Huang**[2], **Philip Lin Huang**[2], **Robert H. Shoemaker**[3], **Sylvia Lee-Huang**[4]*, **Paul T. Wingfield**[1]

**1** Protein Expression Laboratory, NIAMS, NIH, Bethesda, Maryland, United States of America, **2** Department of Medicine, Harvard Medical School and Massachusetts General Hospital, Boston, Massachusetts, United States of America, **3** Chemopreventive Agent Development Research Group, Division of Cancer Prevention, NCI, NIH, Bethesda, Maryland, United States of America, **4** Department of Biochemistry and Molecular Pharmacology, New York University, Grossman School of Medicine, New York, New York, United States of America

⊕ These authors contributed equally to this work.
* wattsn@mail.nih.gov (NW); sylvia.lee-huang@med.nyu.edu (SLH)

**Data Availability Statement:** All assay data are available in the Dryad Repository (DOI: 10.5061/dryad.z08kprrj4).

## Abstract

The continuing emergence of SARS-CoV-2 variants has highlighted the need to identify additional points for viral inhibition. Ribosome inactivating proteins (RIPs), such as MAP30 and Momordin which are derived from bitter melon (*Momordica charantia*), have been found to inhibit a broad range of viruses. MAP30 has been shown to potently inhibit HIV-1 with minimal cytotoxicity. Here we show that MAP30 and Momordin potently inhibit SARS-CoV-2 replication in A549 human lung cells ($IC_{50} \sim 0.2$ μM) with little concomitant cytotoxicity ($CC_{50} \sim 2$ μM). Both viral inhibition and cytotoxicity remain unaltered by appending a C-terminal Tat cell-penetration peptide to either protein. Mutation of tyrosine 70, a key residue in the active site of MAP30, to alanine completely abrogates both viral inhibition and cytotoxicity, indicating the involvement of its RNA N-glycosylase activity. Mutation of lysine 171 and lysine 215, residues corresponding to those in Ricin which when mutated prevented ribosome binding and inactivation, to alanine in MAP30 decreased cytotoxicity ($CC_{50} \sim 10$ μM) but also the viral inhibition ($IC_{50} \sim 1$ μM). Unlike with HIV-1, neither Dexamethasone nor Indomethacin exhibited synergy with MAP30 in the inhibition of SARS-CoV-2. From a structural comparison of the two proteins, one can explain their similar activities despite differences in both their active-sites and ribosome-binding regions. We also note points on the viral genome for potential inhibition by these proteins.

## Introduction

The continuing emergence of SARS-CoV-2 variants capable of eluding monoclonal antibody therapies, conventional and mRNA vaccines, and the human immune response to infection, has highlighted the importance of identifying additional means of intervention.

Ribosome-inactivating proteins (RIPs) have repeatedly been shown to inhibit a broad range of plant and animal viruses including double-stranded DNA viruses, positive-sense single-stranded RNA viruses, negative-sense RNA viruses, and retroviruses, and to therefore have

**Funding:** This research was supported by the Intramural Research Program of the NIH National Institute of Arthritis and Musculoskeletal and Skin Diseases.

**Competing interests:** The authors have declared that no competing interests exist.

therapeutic potential (for reviews see [1–6]; for a critique see [7]). Saporin, an RIP derived from soapwort (*Saponaria officinalis*) has recently been proposed as a therapeutic for SARS-CoV-2, a positive-sense single-stranded RNA virus [8], as has RTAM-PAP1, a fusion protein of the Ricin A-chain and Pokeweed Antiviral Protein isoform 1, obtained from *Ricinus communis* and *Phytolacca americana*, respectively [9].

RIPs are RNA N-glycosylases [EC 3.2.2.22] that depurinate eukaryotic and prokaryotic rRNAs, thereby arresting protein synthesis. Some also possess DNA glycosylase activity. RIPs are structurally related but broadly divided into two classes; Type I RIPs composed of a single polypeptide (A-chain) with glycosylase activity, and Type II RIPs composed of a catalytic A-chain and a carbohydrate-binding (lectin) B-chain joined by a disulfide bond. Type I RIPs are present in many plants used for human consumption including rice, maize, and barley and they are generally considered non-toxic when consumed. Type II RIPs are much more toxic due to presence of the B-chain which facilitates cell entry [6, 10, 11].

The Type I RIPs MAP30 and Momordin (54.5% sequence identity over residues 1–244) (Fig 1), both derived from bitter melon (*Momordica charantia*), a common food and traditional medicine in Asia and Africa, have both antiviral and anticancer properties [10, 12–19]. MAP30 has been shown effective against HIV-1 with 50% inhibitory concentration ($IC_{50}$) values in the sub-nanomolar range and a therapeutic index of over 1000 [12, 20]. At least 20 other RIPs also inhibit HIV-1 [6]. MAP30 has further been reported to synergize with Dexamethasone and Indomethacin in the inhibition of HIV-1, lowering their $IC_{50}$ by more than a 1000-fold to $10^{-7}$ and $10^{-8}$ M, respectively [20, 21]. Dexamethasone, a corticosteroid, interacts with eukaryotic initiation factor 4E (eIF4E) [22] and reduces inflammation and mortality in Covid-19 patients [23]. Indomethacin, a non-steroidal anti-inflammatory, has also proven beneficial for Covid-19 patients [24]. MAP30, like other RIPs, is an RNA N-glycosylase that specifically depurinates the first adenine (A-4324) in the α-sarcin-ricin recognition loop (SRL; CG**A**GAG) of the 28S rRNA of mammalian ribosomes, preventing binding of the elongation factor EF2, mRNA-tRNA translocation, and protein synthesis [25–27]. Precisely how MAP30 engages ribosomes is not yet known. In the case of Ricin, ribosome inactivation first involves the active-site-distal residues R193 and R235 (corresponding to K171 and K215 in MAP30) in engaging the ribosomal stalk P-proteins [28], and then the active-site residue Y80 (Y70 in MAP30) in cleavage of the A-4324 glycosidic bond [10]. An 11-residue peptide (SDDDMGFGLFD) in the C-termini of P-proteins (termed c11-P), normally involved in the binding of elongation factors, has been identified in the binding of Trichosanthin (TCS), an RIP obtained from Chinese cucumber (*Trichosanthes kirilowii)* [29].

Viral inhibition by RIPs may also be due to depurination of other polynucleotides including DNA, poly-A, mRNA, tRNA, and the viral genome [30, 31]. In the case of SARS-CoV-2, this would be a positive-sense single-stranded RNA [32]. As abasic RNAs are resistant to cleavage, the modified 3D structure of the genome may well affect its translation and other interactions in the cell [8]. MAP30 and α-Momocharin, another RIP present in bitter melon, can cause topological inactivation of HIV-1 DNA [33, 34]. Saporin has been proposed to induce apoptosis by various means including DNA damage; PRP1 activation, NAD+ depletion, and mitochondrial damage; rRNA depurination, ribotoxic stress, and MAPK activation; and without rRNA depurination by binding to the cell surface chemokine receptors CCR1, CCR2B, CCR3, CCR4, CCR5, and CXCR4 [8]. More recently, *in silico* docking studies have also predicted direct interactions between RIPs and the SARS-CoV-2 Spike, Mpro, PLpro, RdRp, and E proteins [9]. In addition to the various cellular and viral targets mentioned above, MAP30 may also inhibit SARS-CoV-2 in another way. Specific peptides from both MAP30 and Momordin, that have been shown to reduce hypertension in a rat model and to competitively inhibit angiotensin-converting enzyme (ACE), a homolog of ACE2 [35] and the key receptor for

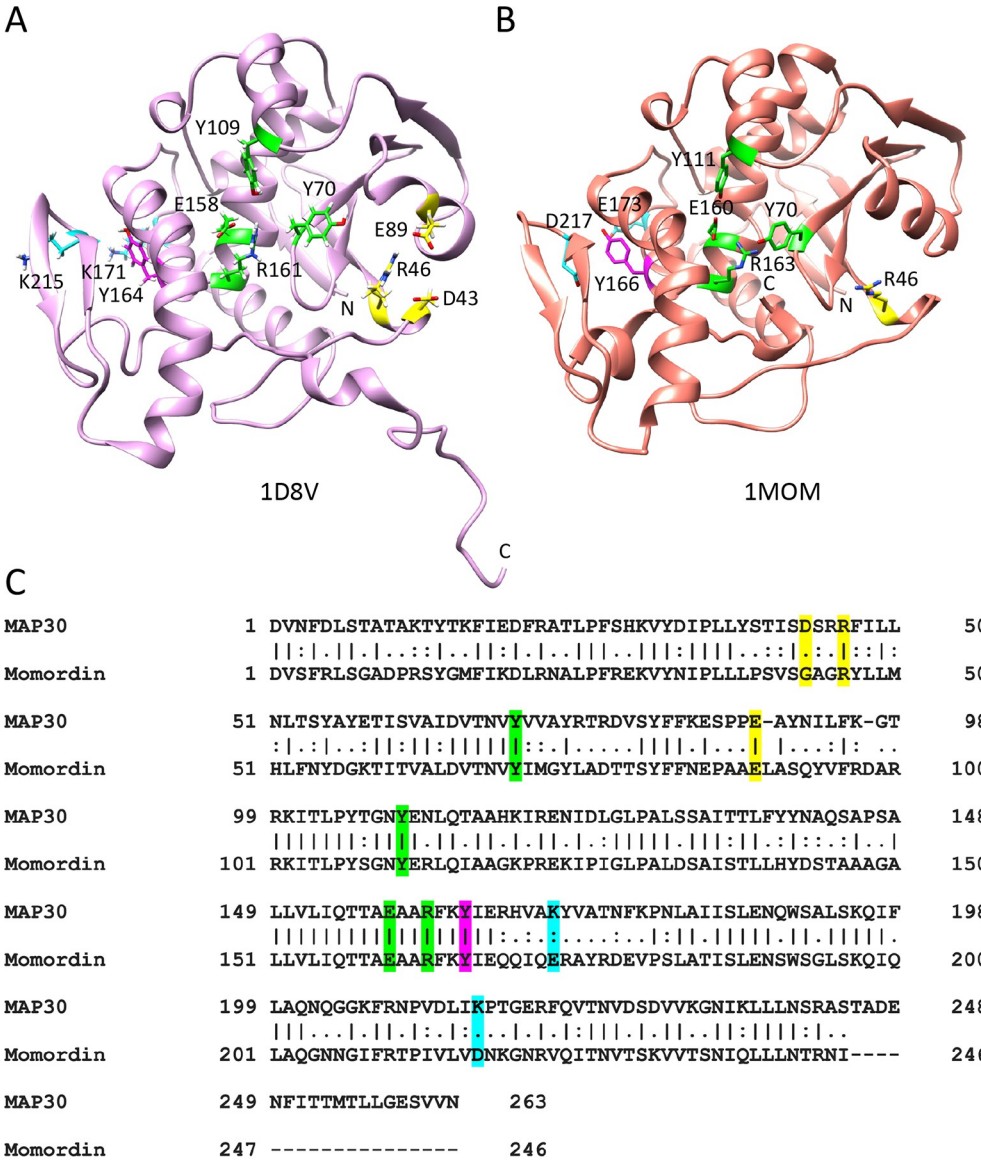

**Fig 1. Structure comparison of MAP30 and Momordin.** (A) MAP30 and (B) Momordin as viewed towards their respective active-sites, with key residues discussed in the text highlighted as follows: catalytic-site (green), $Mn^{+2}$-binding site (yellow), and on the active-site-distal face, the ribosome-binding site (cyan) with the c11-P peptide-interacting Tyr residue (magenta). (C) Sequence alignment of MAP30 and Momordin with residues highlighted as above. Note that the structures for both MAP30 (PDB: 1D8V, 1CF5) and Momordin (PDB: 1MOM, 1AHA) are all missing 23 N-terminal residues. The C-terminal 19 residues are shown in the nuclear magnetic resonance structure of MAP30 (PDB: 1D8V) but not in the X-ray crystal structures of either protein. The N-terminal residues were missing, and C-terminal residues were present, on both proteins employed here.

SARS-CoV-2, have the potential to block viral entry. During our earlier structure determination of MAP30 by means of nuclear magnetic resonance spectroscopy [10] it also became evident that MAP30 is an exceptionally robust protein. As MAP30 is present in a widely consumed food, and as we had also previously found MAP30 to be rather non-toxic, due to an apparent inability to enter uninfected cells [36], these observations all initially suggested that it might be developed as an aerosolized SARS-CoV-2 inhibitor.

In this study, we have assessed MAP30 and Momordin inhibition of SARS-CoV-2 in a nano-luciferase reporter virus (NLRV) assay using A549 human lung cells expressing ACE2. We have assessed both proteins, appending a Tat cell-penetration peptide to each, synergy with Dexamethasone and Indomethacin, and mutating residues in both the active-site and the presumptive ribosome-binding-site of MAP30. We observed similar potent viral inhibition by both proteins with minimal toxicity at the active concentration. Structural analysis of the two proteins can explain their consistently similar activities despite substantial differences in sequence. We also identify sites on the viral genome as potential points for inhibition by these proteins.

## Materials and methods

### Recombinant protein production

MAP30 and variants were all well expressed in *E. coli*. Wild type protein and the mutants MAP30.D43A, MAP30.Y70A, and MAP30.K171A, K215A were purified as previously described using ion-exchange chromatography and gel filtration (S1 Fig) [10]. For the double mutant, some adjustments in pH of ion-exchange columns were required due to a shift in protein isoelectric point. MAP30 with a C-terminal his-tag (MAP30-hist) was purified using Ni-Sepharose chromatography and gel filtration. MAP30 with a C-terminal Tat cell-penetration peptide (GRKKRRQRRRPQ) required urea extraction from bacterial cell extracts and 4 M urea was included in chromatography steps. The urea was removed by step wise dialysis (3, 2, 1 and 0 M urea) in PBS. Momordin and Momordin-Tat (expressed mostly soluble) were purified using the same procedure as MAP30. All proteins were single peaks on gel filtration columns and were concentrated using Amicon Ultra-4, 10,000 NMWL centrifugal filtration units and sterile filtered using Whatman Anotop 10, 0.1 µm filters prior to use. Soluble recombinant human ACE2 (APN01) was provided by Apeiron Biologics.

### Analytical centrifugation

A Beckman Optima XL-I analytical ultracentrifuge, absorption optics, an An-60 Ti rotor and standard double-sector centerpiece cells were used. Sedimentation equilibrium measurements at 20˚C were made at 17,500 rpm for MAP30 and 11,000 rpm for soluble ACE2 receptor with data collection every 4 h to 16 h with overspeed at 40,000 for 4 h to establish a baseline. Partial specific volumes (v-bar), calculated from the amino acid compositions, and solvent densities estimated using the program SEDNTERP (http://www.rasmb.bbri.org/).

### Circular dichroism

Circular dichroism and thermal denaturation measurements were made using a Jasco-715 spectropolarimeter equipped with a PTC-343W1 Peltier-type thermostatic cell holder. Far-UV spectra were recorded at ~ 1 mg/ml protein concentration using 0.02 cm pathlength cells. Temperature response was monitored at 222 nm using a 1-cm path-length cell with a Teflon stopper (Hellma). Cooling circulating water was supplied using a Neslab RTE-100 thermostatic circulator. Proteins (~ 0.1 OD 280 nm) were heated at 1–2˚C per minute with a temperature slope of 20–90˚C. The step resolution was 1˚C, the response time 1 sec, the bandwidth 2 nm, and the sensitivity 100 mdeg. Temperatures at the transition midpoints, i.e., the melting temperature ($T_m$), were estimated from first derivative plots of the melting curves.

### MAP30 interaction with soluble ACE2

MAP30-hist was immobilized on a HiTrap IMAC Sepharose FF column (GE Healthcare) in PBS buffer. Soluble recombinant human ACE2 receptor (APN01), obtained from Apeiron

Biologics, was loaded, and the column was washed with buffer then eluted with PBS plus 0.3 M imidazole. Alternatively, the MAP30-hist and ACE2 were mixed, loaded on the column, washed with buffer then eluted with imidazole. Under both conditions, the ACE receptor was in the wash fractions, indicating no interaction with the immobilized MAP30 protein. In another approach, the MAP30 and ACE2 proteins were mixed and applied to a Superdex 200 gel filtration column (GE Healthcare). The proteins eluted separately and according to their molecular weights 184,000 (dimeric ACE2) and 30,000 (monomeric MAP30) with no interaction complexes detected.

## Molecular dynamics simulations

MD simulations were carried out using the Desmond-Maestro simulation package (Schrödinger Release 2021–4). Protein and ligand were prepared by Protein Preparation Wizard, and the optimized potentials for liquid simulation force field (OPLS_2005) parameters were used in restraint minimization and system building [37]. The system was set up for simulation using a predefined water model (TIP3P) as a solvent. The electrically neutral system for simulation was built with 0.15 M NaCl in 20 Å buffer. The NPT ensemble with 300˚K, and a pressure of 1 bar was applied in the run. The simulation was performed for 250 ns, and the trajectory sampling was done at an interval of 5 ps. The short-range coulombic interactions were analyzed using a cutoff value of 9.0 Å using the short-range method. The smooth particle mesh Ewald method was used for handling long-range coulombic interactions. The stability of MD simulation was monitored by RMSD of the ligand and protein atom positions in time. Hydrogen-bond and salt-bridge formation was analyzed using Visual Molecular Dynamics (VMD) [38].

## Model assessment

Modeled protein-peptide interfaces were examined with PDBePISA [39]. Protein stabilities were calculated with FoldX/Yasara [40]. RNA 2D structure predictions were made with RNA-fold [41]. RNA 3D structure predictions were made with 3dRNA [42]. Molecular illustrations were prepared with UCSF Chimera [43].

## Cell culture

A549 cells expressing ACE2 (obtained from Ralph Baric at UNC) [44] were grown in DMEM high glucose supplemented with 20% (HI) FBS, 1% NEAA, 100 μg/ml Blasticidin and split 1:6 every three days. Blasticidin was removed from the media one passage before using the cells in the assay. On the day of assay, cells were harvested in assay medium (DMEM supplemented with 2% HI FBS, 1% HEPES, 1% Pen/Strep). Total cell number and percent viability determinations were performed using a Luna cell viability analyzer (Logos Biosystems) and trypan blue dye exclusion. Cell viability greater than 95% was required for the cells to be utilized in the assays.

## Sample preparation

The stock solution was serially diluted 2-fold resulting in 10 concentrations in a 384-well plate, each at 6X of the final assay concentration. Dilutions were performed in assay medium. A 5 μL aliquot was taken from each well containing compound and transferred to corresponding wells of the assay plate (384 well Corning #3764). Two plates were prepared for the anti-viral assay and one plate for the cytotoxicity assay. The plates containing samples for the anti-viral assay and the cell suspension were passed into the BSL-3 facility.

## Antiviral assays

Assays were performed applying high-throughput methods in 384-well plates [45, 46].

## A549 nanoluc reporter virus (NLRV) assay

The day of each assay, a pre-titered aliquot of virus (USA_WA1/2020 modified to express nanoluciferase, received from R. Baric at UNC) [47] was removed from the freezer (-80˚C) and allowed to thaw to room temperature in a biological safety cabinet. A working stock of SARS CoV-2 nanoluciferase reporter virus (NLRV) was diluted 1000-fold in media containing 160,000 A549 cells per mL resulting in an MOI of approximately 0.02. Cells and virus were stirred at 200 RPM for at least 10 minutes. A 25 μL aliquot of virus-inoculated cells (4000 A549 cells/well) was added to each well in columns 3–24 of the assay plates. The wells in columns 23–24 did not contain test compounds, only virus-infected cells for the 0% inhibition controls. Prior to virus inoculation, a 25 μL aliquot of cells was added to columns 1–2 of each plate for the cell-only 100% inhibition control (no virus, no compound). After incubating plates at 37˚C/5% $CO_2$ and 90% humidity for 72 hours, 30 μL of NanoGlo (Promega #N1620) was added to each well according to the manufacturer's instructions with the following modifications: the substrate was diluted 1/500 and plates were incubated for 20 minutes prior to the read. Plates were sealed with a clear cover and surface decontaminated prior to removal from the hood. Luminescence was read using a BMG CLARIOstar plate reader (bottom read) to measure luciferase activity as an index of virus titer.

## Cytotoxicity assay

Compound cytotoxicity was assessed in a BSL-2 counter screen as follows. Host cells in media were added in 25 μL aliquots (4,000 cells/well) to each well of assay ready plates prepared with test compounds as described above. Cells only (100% viability) and cells treated with hyamine (Sigma cat# 1622) at 100 μM final concentration (0% viability) served as the high and low signal controls, respectively, for cytotoxic effect in the assay. DMSO was maintained at a constant concentration for all wells as dictated by the dilution factor of stock test compound concentrations. After incubating plates at 37˚C/5% CO2 and 90% humidity for 72 hours, 30 μL Cell Titer-Glo (Promega) was added to each well. Luminescence was read using a BMG PHERAstar plate reader following incubation at room temperature for 10 minutes to measure cell viability.

## Data analysis

For all assays the raw data from plate readers were imported into ActivityBase v.9.7 (IDBS) where values were associated with compound IDs and test concentrations.

Raw signal values were converted to % inhibition by the following formula:

% inhibition = 100 x abs(test compound value–mean value infected cell controls)/abs(mean value uninfected cell controls–mean value infected cell controls).

For the cell viability assay measuring compound cytotoxicity, % cell viability was calculated as follows:

% viability = 100 x (test compound value—mean low signal control)/(mean high signal control–mean low signal control).

$IC_{50}$ and $CC_{50}$ values were calculated from a four-parameter logistic fit of data using the Xlfit module of ActivityBase.

Curated inhibition and cell viability data from Southern Research formatted in MS Excel were imported into OriginPro, Version 2023 (OriginLab Corporation, Northampton, MA,

USA). Data plots were analyzed using non-linear curve fitting (Category = Pharmacology; Function = Dose Response).

## Matrix assay compound preparation

MAP30 and Dexamethasone, and MAP30 and Indomethacin, were tested in a dual combination design in the A549 NLRV assay.

Each combination consisted of seven concentrations of one compound (cmpd B) across a section of the plate with seven concentrations of the second compound (cmpd A) arranged down the same section of the plate. Serial dilutions were performed to produce a 6X concentrations such that a 5 μL transfer into the assay resulted in the final concentrations indicated (30 μL total assay volume). The anti-viral and cytotoxicity assays were performed in parallel.

Concentration response graphs of compound combinations were produced in GraphPad Prism software package.

For synergy analysis, reduced data with associated concentrations and compound ids were imported into https://synergyfinder.fimm.fi/ which applies models for detection of synergistic interactions. The zero-interaction potency (ZIP) model was applied [48].

## Results and discussion

### Protein production and characterization

MAP30 and variants were all well expressed in *E. coli* as soluble proteins and purified by conventional chromatography (S1A Fig). Similarly, for the Momordin proteins, although the expression levels were lower. MAP30 is a very soluble monomeric protein as established by analytical ultracentrifugation (S1B Fig) and all other proteins were also judged monomeric based on their gel filtration behavior. MAP30 and Momordin are α-helix and β-sheet containing proteins with similar structures (Fig 1A and 1B) and this is reflected by qualitatively similar far-UV circular dichroic spectra derived from secondary structure (S1C Fig). MAP30 is a stable protein which was required for structure determinations by NMR where protein is incubated at 40–42˚C for prolonged time periods. The direct thermal stability was determined by helix-coil transition monitored by far-UV circular dichroism; an apparent melting temperate ($T_m$) of 70.5˚C was determined which was likely underestimated due to temperature induced irreversible denaturation (S1D Fig) Although MAP30 would not be classified as a thermostable protein based on the determined $T_m$, the protein has higher than average stability which would be an asset for any potential biomedical formulation [49]. Momordin is also relatively thermostable but less so than MAP30 (S1D Fig).

### MAP30 does not interact with ACE2

As indicated above, specific peptides from MAP30 and Momordin have been shown to reduce hypertension in rats, and to inhibit angiotensin-converting enzyme (ACE) [35], a homologue of ACE2, the main receptor for SARS-CoV-2 [50]. We therefore initially considered the possibility that MAP30 might also interact with ACE2 and block viral entry. Our preliminary *in silico* docking attempts indicated high surface complementarity between two MAP30 monomers and a model of the ACE2 dimer (S2A Fig), with the ACE inhibitory peptides of the proteins reasonably well positioned in the catalytic clefts, further suggesting that MAP30 and Momordin might function as inhibitors of viral entry.

To test for such an interaction between MAP30 and ACE2 we used a recombinant form of human ACE2 (APN01) which we were assaying at the time for tolerability as a potential Covid-19 therapeutic [51]. We first determined the self-association state of each protein by

sedimentation equilibrium analysis and found MAP30 to be monomeric and ACE2 to be dimeric (S2B Fig). The proteins were then mixed and assayed by both gel filtration and Ni-Sepharose chromatography, but no evidence of an interaction was observed (S2C and S2D Fig). This indicated that intact MAP30 does not interact with ACE2 and likely would not function as an inhibitor of viral entry.

## A549 NLRV assays

The results reported below derive from assays performed over a two-year period beginning with the start of the SARS-CoV-2 pandemic. Due to risk and urgency at the time the viral inhibition assays were performed at Southern Research in Birmingham, AL. Over that period, all the assays were replicated at least once, and for wild-type MAP30 numerous times as it was present in every assay. The replicate assays were performed with independent preparations of the proteins and assayed months apart with independent preparations of the cells.

Inhibition of viral replication was tested in a nanoluciferase reporter virus (NLRV) assay using A549 cells expressing ACE2. Both the reporter virus and the cells had been provided to Southern Research by the laboratory of Ralph Baric (UNC) specifically to assay potential SARS-CoV-2 replication inhibitors. A549 cells are commonly used to model the alveolar Type II pulmonary epithelium. They have been used to study various conditions including cancer, allergies, asthma, and respiratory infections. A549 are adenocarcinomic human alveolar basal epithelial cells and therefore high in the expression of LRP1 (https://www.proteinatlas.org/ENSG00000123384-LRP1/tissue), the putative receptor for MAP30 [52]. It was a concern that A549 cells might take up MAP30 merely because they are transformed [14]. However, MAP30 has been tested in a wide range of human cancer cells including T-cell lymphoma, T-cell leukemia, glioblastoma, and breast, lung, liver, cervical, bladder, gastric, ovarian, and kidney cancer cells [12–14, 18, 20, 53, 54], suggesting that the A549 cells might still be a useful model, particularly for virally infected cells.

## Antiviral activity of MAP30 and Momordin

Using the A549 cells we assayed MAP30 and Momordin, each with and without a C-terminally appended Tat cell-penetration peptide, for inhibition of viral replication. There was clear viral inhibition (overall mean $IC_{50}$ = 5.7 μg/ml or ~ 0.2 μM) with little cytotoxicity (overall mean $CC_{50}$ = 72 μg/ml or ~ 2.4 μM) (Fig 2; Table 1, Assay 1). These values are in the ranges observed previously, 0.2–10 μM and 3–30 μM for 50% inhibition and toxicity, respectively, for Type 1 RIPs [6]. The $IC_{50}$ values also compare favorably with Chloroquine in this system (mean $IC_{50}$ = 4 μM). Interestingly, there was no substantial difference between MAP30 and Momordin, with or without the Tat-CPP. This result suggested that the catalytic (RNA N-glycosylase) activity common to both MAP30 and Momordin might be the essential property. This conclusion was strengthened by the observation that almost all of the 17 residues within 4.5 Å of the catalytic residue Y70, in both proteins, are identical and that the residues Y70, Y109, E158, and R161 responsible for this activity in MAP30 [10], and Y70, Y111, E160, and R163 in Momordin [55], are in the same relative positions. It also suggested that cell entry was mediated by factors other than the Tat-CPP, perhaps LRP1 as proposed recently [52].

## MAP30 and Momordin show no synergy with Dexamethasone and Indomethacin

With the observation that MAP30 inhibited SARS-CoV-2 as effectively as Momordin, either one with or without a Tat-CPP, we asked if MAP30 might synergize with Dexamethasone. In parallel, we also tested Indomethacin as we have previously observed that its inhibition of

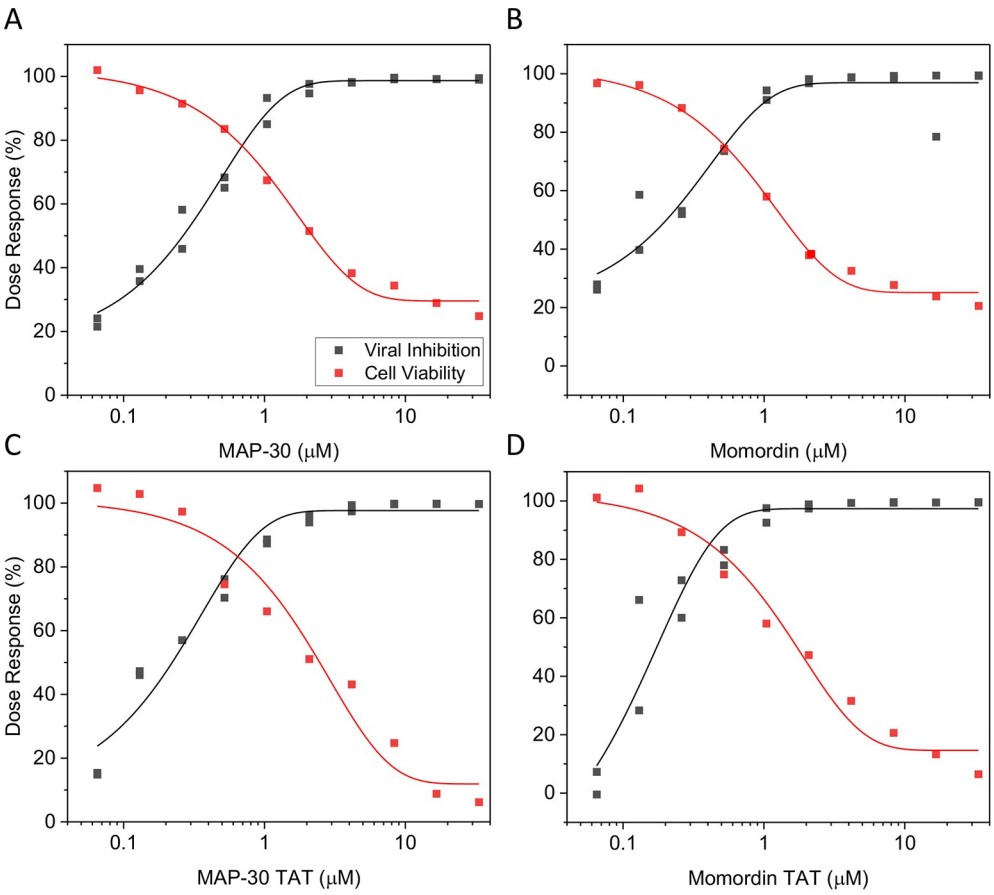

**Fig 2. SARS-CoV-2 inhibition by MAP30 and Momordin.** Viral inhibition and cell viability as a function of concentration of MAP30 and Momordin, either without (A and B) or with (C and D) a C-terminally appended Tat cell penetration peptide, as assessed in the A549 NLRV assay. The results from this and other assays are summarized in Table 1.

HIV-1 is also potentiated by MAP30 [20]. Other than a modest increase in the selectivity index value due to additivity with Dexamethasone, no synergy was observed with either compound, either due to differences between HIV-1 and SARS-CoV-2, or differences between the MT-4 lymphocytes used in the earlier study and the A549 lung cells used here (Table 1, Assay 2).

## Mutational analyses of MAP30 active- and ribosome-binding sites

We also tested whether the RNA N-glycosylase activity is the essential property of MAP30 responsible for viral inhibition by mutating Y70 in the catalytic site. A MAP30.Y70A mutant was tested in the A549 NLRV assay and found to be completely inactive, showing that this activity is central to SARS-CoV-2 inhibition (Fig 3; Table 1, Assay 3).

Having observed relatively potent viral inhibition, we next sought ways to increase the selectivity index. Ricin A-chain protein synthesis inhibition involves its binding to the ribosomal stalk P-proteins followed by depurination of A-4324 in the SRL of the 28S rRNA, and this binding was shown to involve Ricin-A residues R193 and R235 [28, 56]. To reduce or eliminate MAP30 toxicity due to ribosomal inactivation, while retaining the RNA N-glycosylase activity and potentially restricting depurination to viral targets, such as points on the viral genome, we tested the corresponding MAP30.K171, K215 construct. Disappointingly, the

**Table 1. Viral inhibition and cytotoxicity values of MAP30 and Momordin.**

| Assay[a] | Test item | IC$_{50}$ (µg/ml)[b] | CC$_{50}$ (µg/ml)[c] | SI[d] |
|---|---|---|---|---|
| 1 | MAP30 | 6.6 | 100.6 | 15.2 |
| | MAP30-Tat | 5.8 | 72.6 | 12.5 |
| | Momordin | 5.1 | 59.3 | 11.6 |
| | Momordin-Tat | 5.3 | 55.6 | 10.5 |
| | Chloroquine[e] | 1.27 | >10 | >7.9 |
| 2[f] | MAP30 | 5.9 | 50.8 | 8.6 |
| | MAP30 x Dexamethasone[g] | 9.5 | 192.7 | 20.3 |
| | MAP30 x Indomethacin[g] | 6.5 | 53.9 | 8.3 |
| | Chloroquine | 1.07 | >10 | >9.3 |
| 3 | MAP30 | 9.6 | 96.8 | 10.1 |
| | MAP30.Y70A | >1099 | >1099 | 1.0 |
| | Chloroquine | 0.86 | >10 | >11.6 |
| 4 | MAP30 | 7.4 | 101.7 | 13.7 |
| | MAP30.K171A, K215A | 33.0 | 318.6 | 9.7 |
| | Chloroquine | 1.37 | >10 | >7.3 |

[a] All are A549 human lung cancer cell nanoluciferase reporter virus (NLRV) assays.

[b-d] IC$_{50}$, 50% inhibitory concentration; CC$_{50}$, 50% cytotoxicity concentration; SI, selectivity index, SI = CC$_{50}$/IC$_{50}$. The IC$_{50}$ and CC$_{50}$ values given in the text for Assay 1 are the respective overall means for the four proteins.

[e] Three reference compounds (Calpain inhibitor IV, Remdesivir, and Chloroquine) were present in all assays. Chloroquine gave the most consistent data: IC$_{50}$ = 4.1 ± 0.93 µM; CC$_{50}$ >30.00 µM; n = 7.

[f] Matrix format (protein concentration range, 0–80 µg/ml) x (drug concentration range, 0–10 µg/ml).

[g] At highest drug concentration (10 µg/ml).

effect of this double mutation was to shift both the IC$_{50}$ and CC$_{50}$ to higher values with the result that there was essentially no change in the selectivity index (Fig 3; Table 1, Assay 4). As we subsequently found that in Momordin the corresponding residues are acidic, it appeared that the Ricin paradigm may not strictly apply.

## Structural comparison of MAP30 and Momordin

In the viral inhibition assays described above, we compared MAP30 and Momordin because it would provide a natural experiment that might lead to insights into how differences between the two structures can lead to any differences in their selective indices, and how the latter may be increased. However, as MAP30 and Momordin consistently had essentially identical IC$_{50}$ and CC$_{50}$ values in the A549 NLRV assays, and as mutation to Ala of residues K171 and K215 in the putative ribosome binding site was ineffective at increasing the selectivity index of MAP30, we sought to better understand their structures.

Inspection of a set of RIPs varying in sequence identity from ca. 23–60% compared to MAP30, and rendered as either hydrophobicity or charge surfaces, shows them to be different in several ways. For example, the active-site side of MAP30 appears overall to be slightly less hydrophobic and more acidic than that of Momordin (Fig 4A–4C, 4D–4F). When the first structure of MAP30 was determined by NMR (PDB: 1D8V), titration with the HIV-1 LTR identified residues around the active-site involved in DNA binding [10]. Interestingly, titration with Mn$^{+2}$ and Zn$^{+2}$ ions, previously found to be important in buffers for MAP30 and other RIPs to function, led to chemical shift perturbations that indicated the existence of two distinct binding regions for these ions, one on either side of the active-site [10]. However, comparison of the MAP30 and Momordin structures here shows that while both proteins have the DNA

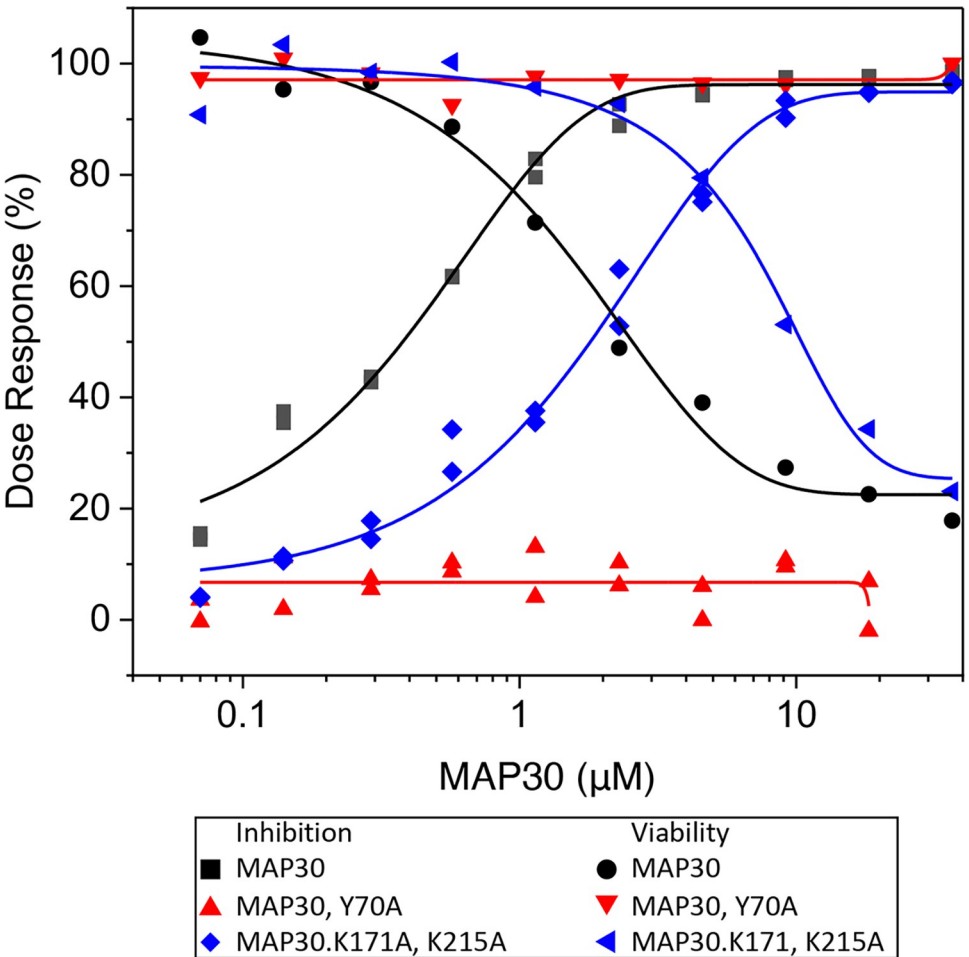

**Fig 3. Mutation of MAP30.** Assessment of SARS-CoV-2 inhibition by MAP30, MAP30.Y70A, and MAP30.K171, K215 in the A549 NLRV assay. The Y70A active-site mutation abrogates both viral inhibition and cytotoxicity. The MAP30.K171, K215 ribosome-binding-site mutations result in an increase of both the $IC_{50}$ and $CC_{50}$ values, while the selectivity index (SI = $CC_{50}/IC_{50}$) remains the same.

and $Zn^{+2}$ binding sites, Momordin does not have either of the D43 or E89 residues constituting part of the $Mn^{+2}$ binding site, although a third residue (R46), which might contribute to the coordination of the ion, is present in both (Fig 1; Fig 4D–4F). In fact, comparison of the other structures in S1 Table shows that all but Saporin do not have the $Mn^{+2}$ binding site.

## MAP30 and Momordin active-site faces

To assess the conservation of residues in and around the RIP active-sites we employed the Evolutionary Trace method which combines phylogenetic and entropic information from multiple sequence alignments to provide a measure of their relative importance [57]. Thus, while the overall sequence identity of MAP30 and Momordin is 54.5%, the top 25% of important residues are 87.6% identical, and they are located closely around the catalytic site (Fig 4G–4I). Neither D43 nor E89 were ranked amongst these residues, suggesting that these two residues and the $Mn^{+2}$ site may not play a role in all MAP30 reactions.

Manual fitting of the SRL (PDB: 430D) into the active-site of Momordin (PDB: 1AHA), with the extra-helical adenine base in the loop aligned to the protein-bound base, and the

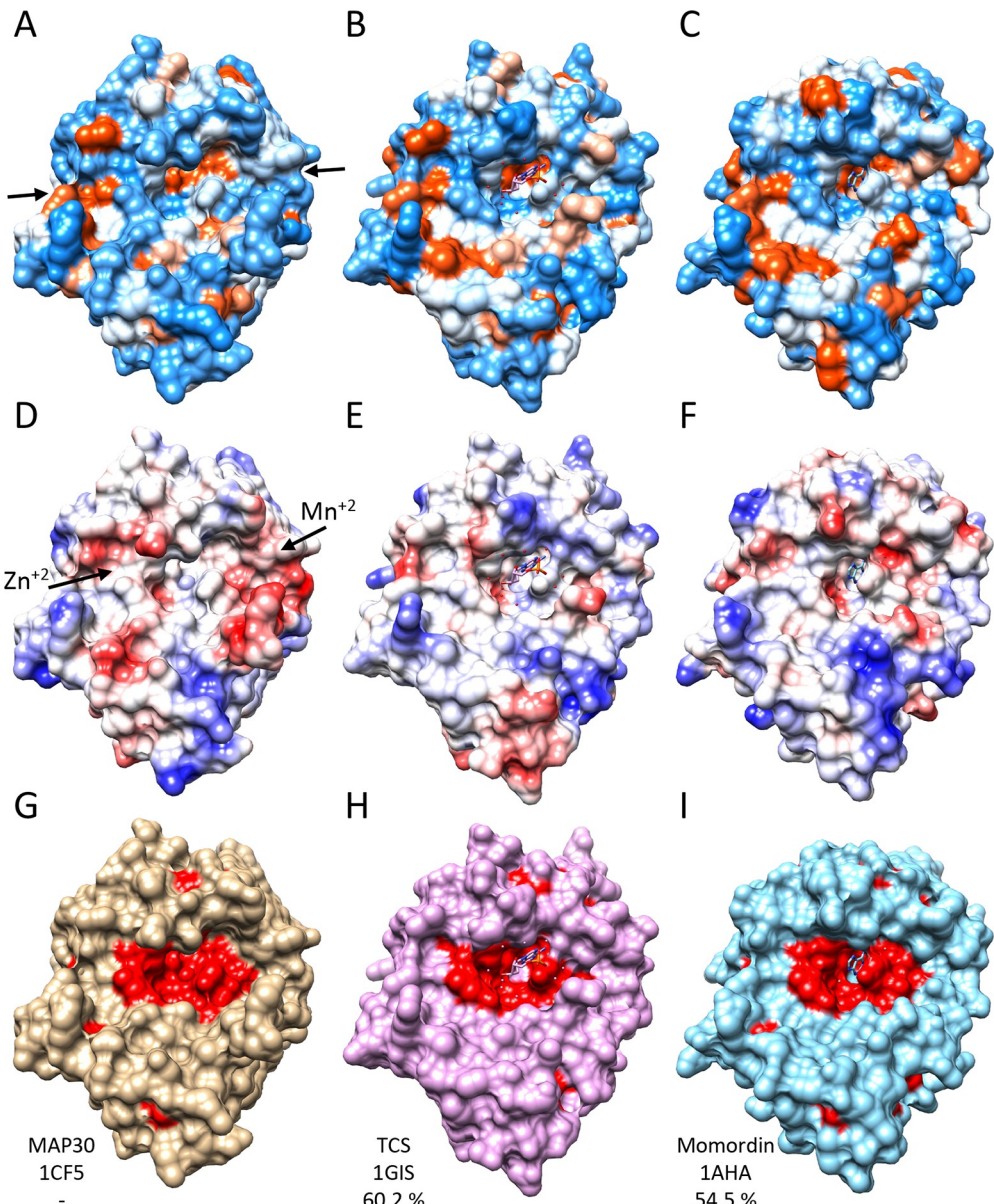

**Fig 4. Comparison of MAP30 and Momordin active-site faces.** (A-C) MAP30, TCS, and Momordin represented as hydrophobicity surfaces (blue, charged; orange, hydrophobic; white, neutral), (D-F) as charge surfaces (blue, basic; red, acidic; white, neutral), and (G-I) with residues scored as important by Evolutionary Trace shown in red. PDB codes and percent sequence identity to MAP30 as indicated. Ricin (PDB: 1APG, 30.5%) and Saporin (PDB: 1QI7, 23.5%) gave similar patterns (not shown). The binding groove with the catalytic site at the center is located between the two arrows in (A). The regions of the previously identified $Mn^{+2}$ and $Zn^{+2}$ binding areas are indicated in (D).

RNA loop rotated to minimize clashes with the protein, gave a visually good fit (S3 Fig). In this placement, the loop extends perpendicularly outward from the active-site without approaching either the $Mn^{+2}$ and $Zn^{+2}$ sites, suggesting that, unlike with the HIV-LTR DNA, these ion binding sites do not play a role in depurination of the RNA, and providing at least a partial explanation for the absence of a difference between the activities of MAP30 and Momordin.

## MAP30 and Momordin active-site-distal faces

MAP30 and Momordin also differ in their active-site-distal surfaces, for example Momordin is more acidic here than MAP30 (Fig 5D and 5E), where, by analogy to TCS [29], they probably interact with the ribosomal P-proteins prior to depurination of A-4324 in the SRL. The X-ray

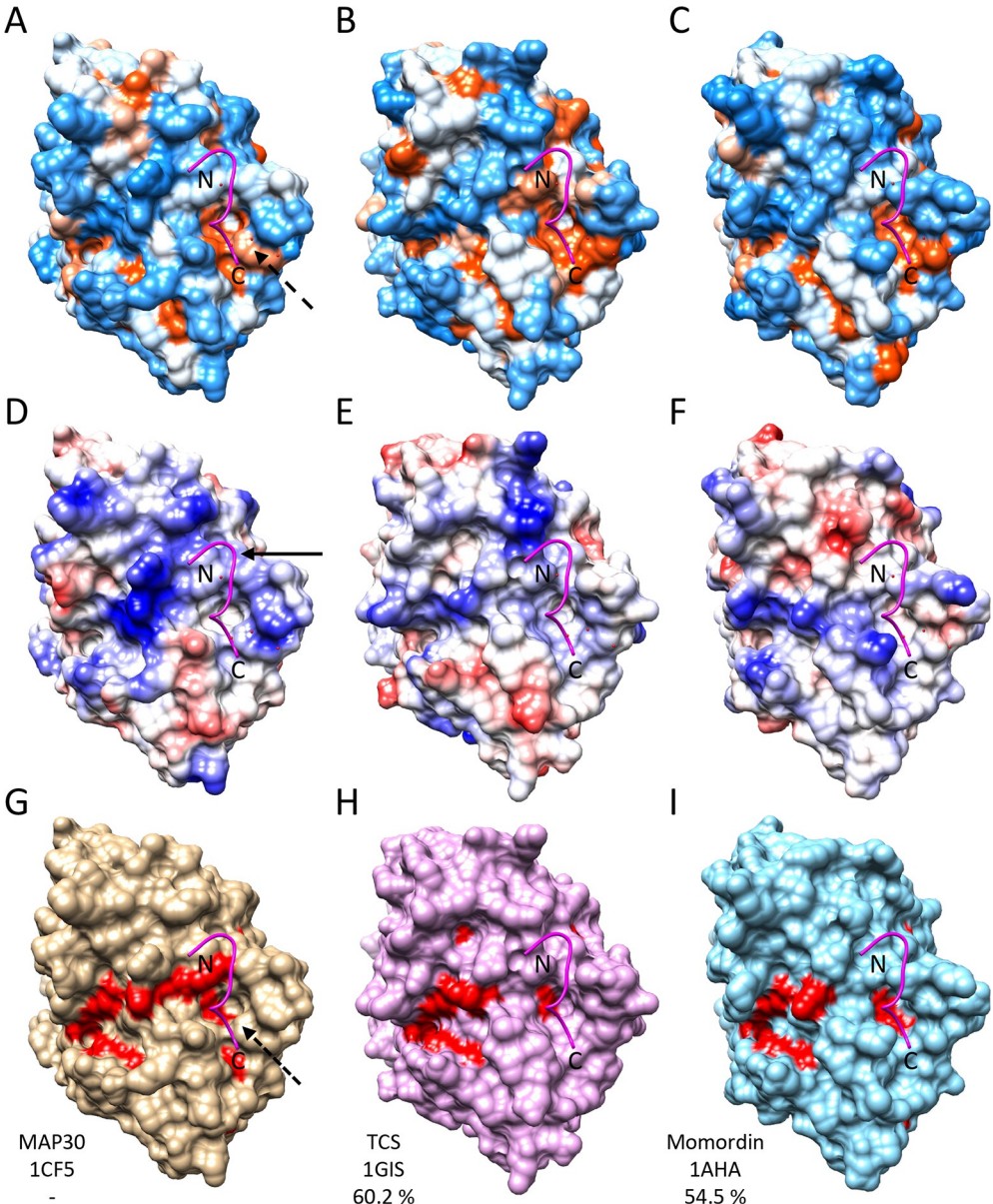

**Fig 5. Comparison of MAP30 and Momordin ribosome-binding-site faces.** (A-C) MAP30, TCS, and Momordin represented as hydrophobicity surfaces (blue, charged; orange, hydrophobic; white, neutral), (D-F) as charge surfaces (blue, basic; red, acidic; white, neutral), and (G-I) with residues scored as important by Evolutionary Trace shown in red. PDB codes and percent sequence identity to MAP30 as indicated. Ricin (PDB: 1APG, 30.5%) and Saporin (PDB: 1QI7, 23.5%) gave similar patterns (not shown). The c11-P peptide bound to TCS, and aligned to MAP30 and Momordin, is shown in magenta. The C-terminal end of the peptide is bound in a hydrophobic groove common to all the proteins (A-C) and the N-terminal end, with acidic resides D2, D3, and D4 (indicated by the arrow in D), likely interacts with a basic region on MAP30 but not the corresponding acidic region on Momordin (D and F). The hydrophobic groove (indicated by a dashed arrow in G), and particularly Y164 in MAP30 and Y166 in Momordin, appear to be important (G and I).

crystal structure of the c11-P peptide (SDDDMGFGLFD) bound to TCS (PDB: 2JDL) when aligned to the crystal structures of MAP30 (PDB: 1CF5) and Momordin (PDB: 1AHA) suggests that the peptide probably binds into a hydrophobic groove in these RIP proteins in a similar manner (Fig 5A–5C). Indeed, for the three modeled complexes, the interaction areas and free energies are all very similar (S2 Table). In MAP30 the acidic N-terminal end of the peptide appears to interact with a basic patch, whereas in Momordin this patch is acidic (Fig 5D and 5E). Highlighting the residues identified with Evolutionary Trace as above (Fig 5G–5I) suggests that the hydrophobic groove is important, potentially so for binding to the ribosomal P-proteins. To investigate this further we performed a molecular dynamics simulation of the binding of the c11-P peptide to both MAP30 and Momordin (S4 Fig).

## Molecular dynamics simulations of MAP30 and Momordin interaction with ribosomal P-protein peptide

In MAP30, the main residue predicted to engage the peptide is Y164, interacting with F10 in replicate simulations 69–72% of the time (S4A and S4B Fig). This involves a π-π face-to-edge interaction. A second set of (charge) interactions involves K215 interacting with D11 and the peptide C-terminus. A third (polar) interaction is between N234 and the L9 carbonyl of the peptide. A fourth set of (charge) interactions involves R167 and K171 with D2, D3, and D4 at the N-terminal end of the peptide.

In Momordin, the main residue predicted to engage the peptide is Y166, interacting with F10 in replicate simulations 37–94% of the time (S4C and S4D Fig). This involves a π-π face-to-edge interaction. A second set of (charge) interactions involves N218, K219, and K231 with D11 and the peptide C-terminus. A third (polar) interaction occurs between S235 and the L9 carbonyl of the peptide. The predicted interactions between MAP30 R167 and K171 and the peptide N-terminal residues D2, D3 and D4 cannot occur in Momordin as the corresponding residues are Q169 and E173. MAP30 R167 and K171, and Momordin Q169 and E173, are also not included in the top 25% important residues, suggesting that they may not always play a significant role in how these RIPs engage the ribosome.

In general, as is evident in the two animations (S1 and S2 Movies), the interaction of the c11-P peptide with Momordin is likely more dynamic than with MAP30, particularly at the N-terminal end which even engages intermittently with N33 over 20 Å away. However, the hydrophobic interaction of the peptide C-terminal region with both proteins appears to be quite stable. These observations of similar interactions at the peptide-binding-site, together with the similarities observed at the active-site, likely explain why the activities of MAP30 and Momordin in the A549 NLRV assays are essentially identical, despite an overall sequence identity of ca. 54% and a clear difference in the $Mn^{+2}$ binding site.

While mutation of MAP30 Y164 (or Momordin Y166) may decrease interaction with the ribosome, it is unlikely to increase the selectivity index. This is suggested by the observation above that the MAP30.K215A mutation did not increase the selectivity index. Also, stability calculations predict that mutating MAP30 Y164 to any of the other 20 amino acids is substantially more disruptive than mutating K215 to any other residue, probably due to the buried location of the former. As Y164 in MAP30 is close to R161, which is directly involved in catalysis, mutating Y164 is likely to be deleterious. Similarly, so for Momordin. For this reason, we considered where else MAP30 and Momordin might exert their antiviral effects.

## A potential new point of SARS-CoV-2 inhibition

RIPs, in addition to their specific RNA N-glycosylase activity on ribosomes, also act to a lesser degree on non-ribosomal RNA and DNA. One such activity depends on the 5' $m^7Gppp$ cap

structure and can lead to cis-depurination throughout the mRNA and may be responsible for the antiviral effects of some RIPs (reviewed in [58]). The positive-sense, single-stranded genome of SARS-CoV-2 (NC_045512.2) has a 5' cap [59] and 77 GAGA motifs out of which four are in regions predicted [41] to form stable hairpins (S5A Fig; S3 Table). These four hairpins are predicted to be more stable than the mean of the 73 others, and No. 2 is significantly more stable (-11.34 kcal/mol vs -4.0 ± 2.1 kcal/mol, n = 73) and it has the highest frequency in its ensemble. The No. 2 hairpin also stands out for other reasons. It has two overlapping copies of the G**A**GA motif of which the first has 6 nucleotides (CG**A**GAG) in common with the SRL, and it is this one that is clearly in the predicted loop. Hairpin No. 2, like No. 3, also has the first adenine of the G**A**GA motif, where the depurination occurs, on the stem-distal side of the loop where it would be readily accessible to depurination, however, it was unclear how representative these 2D predicted loops are, as that of the SRL clearly bears no resemblance to its X-ray crystal structure. To gain a better sense of the hairpins, we performed 3D structure predictions for hairpins No. 2 and No. 3 [42]. Mutual alignment and visual inspection suggest that the 3D predicted structures for hairpin No. 2 more closely resemble the SRL than do those for No. 3 (S5B Fig). Manual docking shows that hairpins related to No. 2 might engage with MAP30 and Momordin as the broad groove in which the active site is located provides some leeway in binding (S3 Fig).

The four hairpins are in ORF1ab with hairpin No. 1 in the coding region for Nsp2, which impairs microRNA-induced silencing by human cells [60] and interferon production [61]. Hairpin No. 2 is in the coding region for Nsp3, an essential component of the viral replication/transcription complex [62]. As it is located just upstream of the coding region for Mpro, the main viral protease, it may affect expression of that as well. Hairpins No. 3 and No. 4 are in the coding region for Nsp13, a helicase that unwinds dsRNA and dsDNA in the 5'-3' direction [63]. There is by now a wealth of information about the many ways in which RIPs, particularly Ricin but also others such as MAP30, affect cells and inhibit viruses. MAP30 and Momordin could conceivably also inhibit SARS-CoV-2 replication by depurinating the genome at these points, particularly at hairpin No. 2.

## Vero E6 CPE assays

In parallel with the above, MAP30 and Momordin were also tested for SARS-CoV-2 inhibition in a Vero E6 cytopathic effect (CPE) reduction assay in a cell line expressing ACE2. No viral inhibition was observed until cytotoxicity occurred, which was at approximately the same concentration as in the A549 NLRV assay. The same result was obtained when a Tat cell-penetration peptide was appended to both proteins, and no synergy was observed with either Dexamethasone or Indomethacin in matrix assays (not shown). These observations suggest that SARS-CoV-2 inhibition by RIPs may be cell line specific and that perhaps in Vero cells some factor involved in viral replication is different from that A549 cells.

## Conclusions

RIPs have a long and notable history as promising antiviral agents. To our knowledge, this is the first report that RIPs such as MAP30 and Momordin inhibit SARS-CoV-2. For both proteins, the inhibition and cytotoxicity values, ca. 0.2 and 2 μM, respectively, are in the ranges previously reported for other RIPs and viruses. MAP30 and Momordin consistently have essentially identical inhibitory activities on SARS-CoV-2 despite distinct differences in their structures at both their active-sites and ribosome-binding-regions. It is not clear from the structures how to substantially improve their selectivity indices. Our structural analyses point to sites on the viral genome where these RIPs may also exert their effects.

## Supporting information

**S1 Fig. MAP30 and Momordin proteins.** (A) SDS-PAGE of MAP30 and Momordin, with and without a C-terminally appended Tat cell penetration peptide, after the final step of column chromatography purification. Standards (Lanes 1 and 6), MAP30 (Lanes 2, 3, 4, and 7), MAP30-Tat (Lane 8), MAP30.Y70A (Lanes 9–12, column fractions), MAP30.K171A, K215A (Lanes 13–16, column fractions), Momordin (Lanes 17 and 18), Momordin-Tat (Lanes 19–22, column fractions). (B) Sedimentation equilibrium analytical ultracentrifugation of MAP30 showing that the protein is monomeric with no tendency for self-association and with a mass corresponding to that predicted from sequence (Fig 1C). (C) Far-UV circular dichroic spectra of MAP30 and Momordin. (D) Thermal denaturation of MAP30 and Momordin monitored by helix–coil transitions at 222 nm. The denaturation of both proteins was irreversible.
(PDF)

**S2 Fig. MAP30 and ACE2 do not interact.** (A) Model of how MAP30 might engage ACE2. Two copies of MAP30 (PDB: 1D8V) fitted onto the ACE2 dimer (PDB: 6M18). Top view (left), and side view (right). Note the apparent shape and charge complementarity. (B) Sedimentation equilibrium analysis of MAP30 (left), and rhACE2 (APN01) (right). MAP30 is monomeric and ACE2 is dimeric. (C) Assay for interaction between MAP30 and soluble rhACE2. Separation of MAP30 and ACE2 by Superdex 200 (left), and Ni-Sepharose (right), chromatography. (D) Analysis by SDS-PAGE of fractions from Superdex 200 (left), and Ni-Sepharose (right), chromatography. There is no evidence of an interaction between MAP30 and ACE2.
(PDF)

**S3 Fig. Model of MAP30-SRL interaction.** (A) Ribbon diagram of MAP30 (PDB: 1D8V) with the residues in the active site colored green, and those involved in $Mn^{+2}$ and $Zn^{+2}$ binding colored yellow and pink, respectively. (B) Ribbon diagram of MAP30 with the SRL (PDB: 430D) modeled to align the adenine in the G**A**GA motif to overlap with the bound adenine in Momordin (PDB: 1AHA) and to avoid clashes with MAP30 (C). (D-F) Orthogonal views of the MAP30-SRL complex model.
(PDF)

**S4 Fig. Interaction of the c11-P peptide with MAP30 and Momordin as modeled by molecular dynamics simulation.** (A and C) Interaction fractions of the peptide with MAP30 and Momordin residues, respectively. (B and D) Schematics of peptide atom interactions with MAP30 and Momordin residues, respectively. Note the large contribution of the hydrophobic interaction of F10 in the peptide with Y164 in MAP30 (occurring 72% of the time during the 250 ns simulation) and to a lesser extent with Y166 in Momordin (37%). Note also, the charge interactions of D2, D3, and D4 in the N-terminal region of the peptide with R167 and K171, and of D11 and the C-terminus of the peptide with K215, in MAP30. Similar interactions occur between the C-terminal region of the peptide, but not the N-terminal region, and Momordin.
(PDF)

**S5 Fig. Points on SARS-CoV-2 genome potentially susceptible to RIP inhibition.** (A) 2D structure predictions for the SRL and the four stem-loop structures in the SARS-CoV-2 genome with the G**A**GA RIP recognition motif located in the loop. In stem-loops No. 2 and No. 3, the adenine in the G**A**GA motif where depurination might occur is in a similar position distal to the stem (indicated by an arrow). (B) Ribbon diagrams of (left) the SRL X-ray crystal structure (PDB: 430D), and (right) the top five 3D structure predictions for stem-loops No. 2

and No. 3 in (A). The adenine in the G**A**GA loop where depurination may occur is highlighted in magenta in each case. (C) Alignment of predicted Hairpin No. 2 with the SRL, with the respective first adenine bases in the common CG**A**GAG motif shown as sticks colored by heteroatom. The bases are extrahelical and essentially superimposable.
(PDF)

**S1 Table. MAP30 and Momordin Mn$^{+2}$ binding sites.**
(PDF)

**S2 Table. Binding parameters of c11-P peptide.**
(PDF)

**S3 Table. Predicted RNA hairpin folding parameters.**
(PDF)

**S1 Movie. Molecular dynamics simulation of MAP30 interaction with c11-P peptide.**
(MOV)

**S2 Movie. Molecular dynamics simulation of Momordin interaction with c11-P peptide.**
(MOV)

## Acknowledgments

All viral inhibition assays were performed at Southern Research, Birmingham, AL. APN01 (soluble recombinant human ACE2) was provided by Apeiron Biologics.

## Author Contributions

**Conceptualization:** Paul L. Huang, Philip Lin Huang, Robert H. Shoemaker, Sylvia Lee-Huang, Paul T. Wingfield.

**Formal analysis:** Norman R. Watts, Elif Eren, Sylvia Lee-Huang, Paul T. Wingfield.

**Funding acquisition:** Paul T. Wingfield.

**Investigation:** Norman R. Watts, Elif Eren, Paul T. Wingfield.

**Project administration:** Paul T. Wingfield.

**Resources:** Ira Palmer, Robert H. Shoemaker, Paul T. Wingfield.

**Supervision:** Sylvia Lee-Huang, Paul T. Wingfield.

**Validation:** Norman R. Watts, Paul T. Wingfield.

**Visualization:** Norman R. Watts, Elif Eren, Paul T. Wingfield.

**Writing – original draft:** Norman R. Watts.

**Writing – review & editing:** Norman R. Watts, Paul T. Wingfield.

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
