## [Decision Letter · Decision Letter 0]

27 Feb 2023

PONE-D-23-02011The ribosome-inactivating protein MAP30 inhibits SARS-CoV-2PLOS ONE

Dear Dr. Watts,

Thank you for submitting your manuscript to PLOS ONE. After careful consideration, we feel that it has merit but does not fully meet PLOS ONE’s publication criteria as it currently stands. Therefore, we invite you to submit a revised version of the manuscript that addresses the points raised during the review process. In your revised version, please address the constructive comments of the two reviewers as fully as possible.

We look forward to receiving your revised manuscript.

Kind regards,

Israel Silman

Academic Editor

PLOS ONE

Journal Requirements:

"This research was supported by the Intramural Research Program of the NIH National Institute of Arthritis and Musculoskeletal and Skin Diseases."

4. Please upload a copy of Supporting File 1 (S1 File) which you refer to in your text on page 10.

Reviewers' comments:

Reviewer's Responses to Questions

**Comments to the Author**

1. Is the manuscript technically sound, and do the data support the conclusions?

Reviewer #1: Partly

Reviewer #2: Yes

2. Has the statistical analysis been performed appropriately and rigorously? 

Reviewer #1: Yes

Reviewer #2: Yes

3. Have the authors made all data underlying the findings in their manuscript fully available?

Reviewer #1: Yes

Reviewer #2: Yes

4. Is the manuscript presented in an intelligible fashion and written in standard English?

Reviewer #1: Yes

Reviewer #2: Yes

5. Review Comments to the Author

Reviewer #1: In the manuscript The ribosome-inactivating protein MAP30 inhibits SARS-CoV-2 Watts et al. tested the anti-SARS-CoV-2 activities of the MAP30 protein, which is a well-studied ribosome-inactivating protein, and its close homolog, Momordin. Overall, the authors have presented a comprehensive work of scientific values. However, there are major concerns which the authors should address before the paper can be accepted.

(1) The authors should justify the use of A549-ACE2 instead of other non-cancerous ACE2-expressing cell lines for antiviral experiments. A549 has been found to be sensitive to MAP30 treatment (see Mol Med Rep 2015, 11(5):3553-3558 and other articles). From the data it seems that cell viability and antiviral effect are (reversely) correlated with each other (Fig 2). While MAP30 mutants reduced cytotoxicity, antiviral effect was also abolished. Further, the authors also mentioned (without data shown) that the protein exhibited no effect on Vero E6 CPE assay. All these suggested a possibility, which is of concern, that the reduced viral growth may be simply due to sub-optimal cellular conditions in the presence of MAP30.

(2) Further to (1), despite the authors have showed no interaction between ACE2 and MAP30, there seems a gap between the presented data and the conclusion that the observed antiviral activity was due to RNA N-glycosidase activity. At least a single-cycle time-of-addition assay should be performed for further support.

(3) Structural comparison between MAP30 and momordin, and MAP30/momordin binding to C-11 peptide are themselves interesting. However, they currently look like a standalone portion in the manuscript. It is suggested that the authors rewrite the relevant paragraphs with more emphasis on how their findings and observations are related to anti-SARS-CoV-2 activity.

(4) Paragraph 429-447 suggested that MAP30 may depurinate the viral RNA at hairpin regions. This proposition is exciting. However the evidence presented in the manuscript was weak. The authors could have further demonstrated how the viral RNA hairpin resembles the SRL (e.g. by structural alignment). Besides, Fig S3 only shows how SRL is docked to MAP30. The authors’ claim can be further supported by docking the hairpin to MAP30. Experimental evidence would also be necessary (e.g., can the authors perform a depurination assay using the SARS-CoV-2 hairpin?)

Minor points:

Line 247: Clarify the sentence “Compounds were run in anti-viral and cytotoxicity assays when tested.”

Throughout the manuscript: Not appropriate to treat amino acids as proper nouns

Reviewer #2: RIPs have long been recognized as antiviral proteins in both plants and animals, but the mechanism responsible for this activity continues to be the subject of intense research today. In this paper, the authors mainly report that MAP30 and momordin inhibit SARS-CoV-2 replication in human A549 lung cells. This in itself is an important finding but in addition the authors try to explain how these RIPs carry out this inhibition. They study if MAP30 could interact with ACE2 and block virus entry, then by mutating key residues of MAP30 show the involvement of its RNA N-glycosylase activity and consider the possibility that RIPs could act on viral RNA by showing points on the viral genome for potential inhibition. An interesting structural comparison (of the ribosomal binding sites, ion binding sites…) of MAP30 and momordin is also reported in this work. Overall, the authors contribute to the elucidation of the structure-activity relationships of RIPs and demonstrate the potential of RIPs for the treatment of virus-related diseases.

The studies are designed and conducted in a logical manner and all conclusions match the data presented. The images and the results in general are well presented. However, I found it difficult to read the manuscript in the section "results and discussion". The headings of the subsections do not always specify the content and in some cases the subsections are too long. I recommend making more sections indicating what is to be studied in each case.

Minor comments

-Most of the work in the manuscript has been performed for both MAP30 and momordin and clearly momordin also inhibits SARS-CoV-2 replication. However, only the antiviral activity of MAP30 appears in the title “The ribosome-inactivating protein MAP30 inhibits SARS-CoV-2”. The authors should explain this.

-Line 53: RIPs are rRNA N-glycosylases (EC 3.2.2.22) that catalyze the elimination of a specific adenine (A4324 in rat ribosomes, or the equivalent in other organisms) located in the sarcin-ricin loop (SRL) of animal 28S ribosomal RNA.

Lines 53, 32, 55, 70, 315, 329, 338, 430 Change glycosidase by glycosylase.

-Line 311- . Looking at the values shown on line 311 for "viral inhibition (mean IC50 =5.7...) with low cytotoxicity (mean CC50=72...)" they do not match the values shown in Table 1. Are the values in the text for MAP30, momordin or with the TAT peptides? Are the values in Table 1, mean values or how were the values obtained? Table 1 should be better explained.

-I recommend including Figure S5 in the manuscript to illustrate the potential points of SARS-CoV-2 inhibition.

-Legend of Fig S3. Model of MAP30-SRL interaction. (D-E) Orthogonal views of the MAP30-SRL complex. Panel F should be mentioned in the legend.

6. PLOS authors have the option to publish the peer review history of their article (what does this mean?). If published, this will include your full peer review and any attached files.

Reviewer #1: No

Reviewer #2: No

---

## [Author Response · Author response to Decision Letter 0]

30 Mar 2023

Response To Reviewers

Reviewer #1: In the manuscript The ribosome-inactivating protein MAP30 inhibits SARS-CoV-2 Watts et al. tested the anti-SARS-CoV-2 activities of the MAP30 protein, which is a well-studied ribosome-inactivating protein, and its close homolog, Momordin. Overall, the authors have presented a comprehensive work of scientific values. However, there are major concerns which the authors should address before the paper can be accepted.

(1) The authors should justify the use of A549-ACE2 instead of other non-cancerous ACE2-expressing cell lines for antiviral experiments. A549 has been found to be sensitive to MAP30 treatment (see Mol Med Rep 2015, 11(5):3553-3558 and other articles). From the data it seems that cell viability and antiviral effect are (reversely) correlated with each other (Fig 2). While MAP30 mutants reduced cytotoxicity, antiviral effect was also abolished. Further, the authors also mentioned (without data shown) that the protein exhibited no effect on Vero E6 CPE assay. All these suggested a possibility, which is of concern, that the reduced viral growth may be simply due to sub-optimal cellular conditions in the presence of MAP30.

Before addressing the Reviewer’s concerns, we would like to thank them for their thoughtful comments and helpful recommendations. We hope that the detailed explanations below, and the changes made to the manuscript, will address all their concerns.

The Reviewer raises an important issue. In fact, they raise two broad but interrelated matters; first, the justification for using the A549-ACE2 cells instead of another cell line, and second, the validity of the results eventually obtained from that assay. We have discussed these same questions amongst ourselves at considerable length. We have also had discussions with cell biologist colleagues and with the original developers of the two assays used. There were several considerations, and we will here attempt to address them, not in order of importance but rather for clarity of explanation. We will begin with a brief historical perspective.

In the early days of the pandemic, when the world was in state of panic, there was very little information about the virus, and all possible ways to contain it were being considered. Remarkably, this state of urgency has now been almost forgotten just three years later. As we had previously found that MAP30 potently inhibited HIV-1, and then solved its structure, and therefore knew how to express and purify it, we considered testing if it might also inhibit SARS-CoV-2. The opportunities for testing at that time were extremely limited, even at the NIH. BSL3 facilities and personnel training could not just be implemented immediately, and so we evaluated three NIH-validated testing facilities, in parallel. One proved unable, the second after a long delay produced false negative results, and the third rose to the occasion. That third provider was Southern Research, and with neither time nor facilities to isolate our own cell lines and devise new assays we, like many others, employed the assays that Southern could provide. Even then, Southern struggled early on but quickly established the high-throughput screening capacity that we later found gave impressively reproducible results. Much later, the second provider was also able to obtain positive results with another unrelated and highly effective SARS-CoV-2 inhibitor that we had developed in parallel (PMID: 35816490). These results confirmed exactly those of Southern, giving us even greater confidence in the data from this provider. 

Initially, we employed solely the Vero E6 CPE reduction assay. Only later did we turn to the A549 NLRV assay and do almost everything all over again. The reason for switching from the Vero assays to the A549 assays was that we observed no viral inhibition with the former, except what was merely due to a decrease in viability. We will return to this point in greater detail further below. In an earlier version of the manuscript, we detailed all the Vero results, and all the A549 results, both in equal detail, but then decided to omit the Vero results as they were all “negative”. We did however briefly mention the Vero assay results at the end of the submitted manuscript for the purpose of full disclosure. Quite aside from the MAP30 and Momordin results described in the manuscript, we also assayed numerous peptides from them, both linear and specifically designed cyclic ones. As described in the manuscript, peptides from both MAP30 and Momordin had previously been shown by others to inhibit ACE2, and our molecular modeling attempts in the early days of the pandemic, as the first spike and receptor binding domain (RBD) structures were coming out, suggested that the peptides might block viral entry. All these peptides gave negative results and, much like the full-length-RIP Vero assay results, they are not included in the final manuscript. Thus, a far larger number of potential inhibitors were assayed than are discussed in the submitted manuscript.

The Vero assays were attractive to us because, as the Reviewer points out, a non-cancerous cell line would be desirable in some ways. Unfortunately, essentially all assays at facilities such as Southern are performed with common immortal cell lines such as Hela, HepG-2, HEK293, etc. This is for practical reasons including maintenance and reproducibility over long periods. Also, in a brief survey of the literature, we have found over 20 different human cancer cell lines used to study MAP30 and other RIPs, illustrating that this is an accepted way to study these proteins. The only non-cancerous cells that we have found used with RIPs in the literature are human spermatocytes. Unlike the A549 human lung cells which are cancerous, the Vero African green monkey kidney cells are considered continuous, i.e., they can be passaged without becoming senescent. Vero cells are aneuploid and can, after hundreds of passages, produce cancer-like nodules in nude mice that then recede again. In this limited sense the Vero cell assays might have been preferable to A549 cell assays, but unfortunately, they did not “work”. By this we mean that the inhibition curve was simply the inverse of the viability curve as shown in Fig. 1 (below). This result was obtained numerous times and eventually led us to use the A549 assay. It is worth noting that quite aside from their aneuploid state, the Vero cells are still not quite normal cells because, just like the A549 cells, they are engineered to express ACE2.

Fig. 1. Test of MAP30 and MAP30-Tat in the Vero E6 CPE reduction assay. Shown is a cropped screen shot from one Southern Research activity report. As can be seen, the percent inhibition curve starts to rise at about 100 µg/ml of RIP as loss of cell viability takes over. This response was highly reproducible for both MAP30 and Momordin. We would consider this an example of the situation the Reviewer describes when they suggest that the antiviral effect may simply be due to sub-optimal cellular conditions. This response is quite distinct from that obtained with the A549-NLRV assay where viral inhibition occurs at concentrations one log lower than cytotoxicity (Fig. 2 in the manuscript).

The Reviewer points out that the A549 cells have been found to be sensitive to MAP30 and other RIPs. This is true and we were quite aware of it. It is often thought that MAP30, and likely other Type I RIPs (i.e., without benefit of the B-chain), cannot enter normal cells, but that it can enter cancerous and virally infected cells as the cell membranes have been altered. This makes at least some sense as some RIPs are considered protective antivirals. While the details of this are still far from sorted out in the literature, the premise that normal cells are resistant to MAP30, while virally infected ones are more susceptible, would be of great value were it to be used as an antiviral agent.

This brings us back to the Reviewer’s second question, about the validity of the A549 assay results. Whereas in the Vero assays, where the inhibition curve was the inverse of the viability curve, in the A549 assays there is a log difference between the IC50 and CC50, i.e., a selectivity index (SI) of 10. At cross-over the viral inhibition is ca. 80%. We obtained this result regardless of RIP type, Tat-CPP addition, presence of Dexamethasone or Indomethacin, or mutations to the ribosome interacting face. In some assays we observed a slightly higher Hill slope, but this was not sufficiently high or reproducible to be reported. While the limited selectivity index, and our inability to increase it, was a disappointment, it should be born in mind that some cancer therapeutics began with selectivity indices of 2.

We hope that we have justified the use of the A549 assays to the Reviewer, and that we have convinced them of the validity of the results. It does not seem to us that all these details need to be included in the manuscript, aside from what is already provided in the Results and Discussion. Mentioning at the end that the Vero CPE assay was not suitable for our purpose may save other researchers from trying to use it in this way. Alternatively, they may wish to determine why MAP30 and Momordin are cytotoxic to Vero cells (suggesting they do enter the cells and inactivate the ribosomes) yet they do not inactivate the virus. In other words, what is it about Vero (aneuploid monkey) cells that apparently is somewhat protective of SARS-CoV-2 against RIPs? This is a fascinating question for another study, and another group.

(2) Further to (1), despite the authors have showed no interaction between ACE2 and MAP30, there seems a gap between the presented data and the conclusion that the observed antiviral activity was due to RNA N-glycosidase activity. At least a single-cycle time-of-addition assay should be performed for further support.

We did not wish to imply that a lack of interaction between MAP30 and ACE2 leads to the conclusion that viral inhibition is due its RNA N-glycosidase activity. As summarized in the Introduction, a previous report that peptides from both MAP30 and Mormordin inhibit ACE2 suggested to us that there might be an interaction between MAP30 and ACE2, and our initial molecular modeling efforts supported this. We merely eliminated that as a possible cause of viral inhibition. That the RNA N-glycosidase activity may be involved in viral inhibition has long precedent in the literature, and it was confirmed by our mutagenesis of the MAP30 active site. However, where that activity occurs is unknown. It likely takes place at several sites, including the ribosome and the viral genome. As a typical eukaryotic cell has some 10 million ribosomes, but certainly not that many viral genomes, the cell can potentially recover after viral inactivation. Our A549 NLRV assays provide clear and highly reproducible evidence of viral inhibition by these RIPs. Unlike a small molecule protease inhibitor or capsid assembly inhibitor, RIPs exert their effects at numerous points (summarized in the Introduction (Lines 84-94)). Therefore, we do not think that the proposed time-of-addition assay would resolve these points. We have however inquired with Southern, and it is not an assay that they provide, though they indicated that it might be developed as a custom service.

(3) Structural comparison between MAP30 and momordin, and MAP30/momordin binding to C-11 peptide are themselves interesting. However, they currently look like a standalone portion in the manuscript. It is suggested that the authors rewrite the relevant paragraphs with more emphasis on how their findings and observations are related to anti-SARS-CoV-2 activity.

The Reviewer raises two related issues; the seeming standalone nature of the molecular dynamics simulation, and the how the results from that pertain to anti-SARS-CoV-2 activity. As indicated in the text, comparing two different RIPs provided a natural experiment to see which inhibited SARS-CoV-2 better. This might indicate which one to pursue further, and perhaps also how to increase the therapeutic index of that one. To our surprise, both proteins had essentially identical activities despite an almost 50% difference in sequence, distinct structural differences in the vicinity of the active site, distinct structural differences in the ribosome binding site, and no change in the selectivity index upon mutation of residues in the latter previously identified as important in ribosome binding in another RIP (Ricin). All these observations of the static structures lead us to consider them dynamically, however, that may not have been clear in the text, giving the impression that the simulation is standalone. The molecular dynamics simulation demonstrated similarly stable interactions with the ribosomal c11-P peptide despite differences in the details of binding, particularly at the N-terminal end of the peptide. In other words, the simulation indicated two different, partially overlapping, sets of interactions between the peptide and the two proteins (each different from that with TCS, as described in the literature). This would not have been learned from a static analysis. We have now added several section headings to the Results and Discussion that make these different points more obvious and why we therefore performed a molecular dynamics simulation (Lines 309, 324, 333, 373, 390, and 404).

Regarding the Reviewer’s second point, about how the results pertain to anti-SARS-CoV-2 activity. The generic name ‘Ribosome Inactivating Protein’ implies that this activity is what causes viral inhibition. Yet, surprisingly, even two simultaneous rational mutations in the ribosome binding site had relatively little effect on inhibition. This was in part explained by the molecular dynamics simulations. Instead, the most significant change in activity (complete inactivation) was caused by mutation of the active site, not the ribosome binding site. As stated in the manuscript, we therefore considered where else the N-glycosidase activity might inhibit the virus, and based on sequence analysis and molecular modeling, we predict that it may be the viral RNA genome. Again, this shows how the molecular dynamics simulations are related to understanding the anti-SARS-CoV-2 activity of these RIPs.

(4) Paragraph 429-447 suggested that MAP30 may depurinate the viral RNA at hairpin regions. This proposition is exciting. However the evidence presented in the manuscript was weak. The authors could have further demonstrated how the viral RNA hairpin resembles the SRL (e.g. by structural alignment). Besides, Fig S3 only shows how SRL is docked to MAP30. The authors’ claim can be further supported by docking the hairpin to MAP30. Experimental evidence would also be necessary (e.g., can the authors perform a depurination assay using the SARS-CoV-2 hairpin?)

Due to revisions, paragraph 429-447 is now 438-456. The objective of our study was to determine if MAP30 and Momordin can inhibit SARS-CoV-2. To this end we made six different proteins, assayed them in two cell lines, tested for interactions with ACE2, tested for synergy with two drugs, and performed two molecular dynamics simulations. We have clearly shown that both proteins can inhibit SARS-CoV-2 (SI = 10). Given that viral inhibition was distinct from toxicity, we then also considered where this inhibition might occur. Based on sequence analysis and molecular modeling we propose specific hairpins in the viral genome.

While the Reviewer thinks this is exciting, they feel that we did not make a strong case and note that Fig S3 only shows how the SRL is docked to MAP30. That is because this figure only serves to compare the MAP30 and Momordin active-site faces, and their interactions with the SRL. At this point in the Results and Discussion we have not yet introduced the reader to the concept of RIPs potentially targeting the viral genome. That is illustrated in Fig S5. However, we have taken the Reviewer’s advice and performed a structural alignment of Hairpin No. 2 with the SRL. That is now shown in Fig S5, C. As in the X-ray crystal structure of the SRL (PDB: 430D), the first adenine base in the shared CGAGAG motif in the loop is flipped out into an extrahelical position where it would be susceptible to enzymatic depurination. This high structural similarity is striking and suggests that MAP30 may also depurinate this hairpin in the viral genome. We consider this a prediction rather than a conclusion, and something that others may want to investigate. That is why we have put it in the Supplementary Information. Note: in addition to the 3dRNA structure predictions described in the manuscript, we have now also performed a more focused prediction on Hairpin No. 2 with FARFAR2. The results are essentially the same, giving us further confidence in our prediction (not shown).

Minor points:

Line 247: Clarify the sentence “Compounds were run in anti-viral and cytotoxicity assays when tested.”

We merely intended to say that the inhibition and viability assays were performed at the same time, rather than at different times. We have changed the wording and moved the sentence to the end of the paragraph, where it should logically be (Line 248-249).

Throughout the manuscript: Not appropriate to treat amino acids as proper nouns

This has now been corrected throughout the manuscript (Lines 30, 31, 32, 33, and 34).

 

Reviewer #2: RIPs have long been recognized as antiviral proteins in both plants and animals, but the mechanism responsible for this activity continues to be the subject of intense research today. In this paper, the authors mainly report that MAP30 and momordin inhibit SARS-CoV-2 replication in human A549 lung cells. This in itself is an important finding but in addition the authors try to explain how these RIPs carry out this inhibition. They study if MAP30 could interact with ACE2 and block virus entry, then by mutating key residues of MAP30 show the involvement of its RNA N-glycosylase activity and consider the possibility that RIPs could act on viral RNA by showing points on the viral genome for potential inhibition. An interesting structural comparison (of the ribosomal binding sites, ion binding sites…) of MAP30 and momordin is also reported in this work. Overall, the authors contribute to the elucidation of the structure-activity relationships of RIPs and demonstrate the potential of RIPs for the treatment of virus-related diseases.

The studies are designed and conducted in a logical manner and all conclusions match the data presented. The images and the results in general are well presented. However, I found it difficult to read the manuscript in the section "results and discussion". The headings of the subsections do not always specify the content and in some cases the subsections are too long. I recommend making more sections indicating what is to be studied in each case.

First, we wish to thank the Reviewer for their thoughtful comments and helpful recommendations. We hope that the detailed explanations below, and the changes made to the manuscript, will address all their concerns.

The Reviewer has made a good recommendation about the need for more section headings. We were too familiar with the text and could no longer see it from a new reader’s point of view. We have now added more section headings in the Results and Discussion (Lines 309, 324, 333, 373, 390, and 404).

Minor comments

-Most of the work in the manuscript has been performed for both MAP30 and momordin and clearly momordin also inhibits SARS-CoV-2 replication. However, only the antiviral activity of MAP30 appears in the title “The ribosome-inactivating protein MAP30 inhibits SARS-CoV-2”. The authors should explain this.

The Reviewer is right, we only mentioned MAP30 in the title. This came about because MAP30 was the protein foremost in our minds, as we have studied it the longest, determined the first structure, and demonstrated its efficacy against HIV. We have now included both proteins in the title (Line 2).

-Line 53: RIPs are rRNA N-glycosylases (EC 3.2.2.22) that catalyze the elimination of a specific adenine (A4324 in rat ribosomes, or the equivalent in other organisms) located in the sarcin-ricin loop (SRL) of animal 28S ribosomal RNA. Lines 53, 32, 55, 70, 315, 329, 338, 430 Change glycosidase by glycosylase.

Glycosylase (EC 3.2) is a broad term that comprises two types of glycosidases: O-(and S-) glycosidases (EC 3.2.1) and N-glycosidases (EC 3.2.2). Therefore N-glycosidase is the more specific term. RIPs hydrolyse the glycosidic bond between a nitrogenous base and a sugar, therefore the term N-glycosidase is more appropriate. Also, the term glycosidase has long been used for RIPs (PMID: 10571185, PMID: 7568024, PMID: 7527556, PMID: 3288622, PMID: 3226909, PMID: 3036799, PMID: 9659388, PMID: 3276522, PMID: 2642481, PMID: 15123667, PMID: 26008228, PMID: 35969718, PMID: 9403966). However, we have made the requested changes throughout the text (Lines 32, 53, 55, 70, 317, 334, 343, and 439).

-Line 311- . Looking at the values shown on line 311 for "viral inhibition (mean IC50 =5.7...) with low cytotoxicity (mean CC50=72...)" they do not match the values shown in Table 1. Are the values in the text for MAP30, momordin or with the TAT peptides? Are the values in Table 1, mean values or how were the values obtained? Table 1 should be better explained.

The Reviewer raises two points; that the inhibition values in the text and Table 1 do not agree, and that Table 1 should be better explained. The Reviewer is right, the values given in the text (in the section now titled “Antiviral activity of MAP30 and Momordin”) do not match those in Table 1. They are average values, as was indicated, and as explained here. Altogether, we performed almost forty assays (not counting of course the internal replicates, typically duplicates but in some instances quadruplicates, nor the 8-fold internal multiplicity of each of the eight synergy assays). Half of these involved the A549 NLRV assay, and half the Vero E6 CPE assay. As it turned out that the Vero assay was not suitable for our purpose, we mention it only briefly at the end of the Results and Discussion. As the 6 proteins were prepared at different times over the course of the two-year study, they were assayed in different combinations at different times. All assays were replicated at least once, sometimes months apart, but MAP30 was present in all runs. The results of one run where MAP30 and Momordin, each with and without a Tat-CPP, were assayed side-by-side is shown in Fig. 1 (below). It illustrates the remarkably similar activities of the four proteins. To reduce the many assays to a comprehensible level, we grouped them by type as “Assay 1”, “Assay 2”, etc. in Table 1, although in addition to the assays shown below the proteins were also assayed in other runs. The respective IC50 and CC50 values for the four proteins are so similar in all the assays that we just give the overall means for these four proteins in the text which the Reviewer refers to. To make this clearer for readers, we have now changed the text to "There was clear viral inhibition (overall mean IC50 = 5.7 µg/ml or ~ 0.2 µM) with little cytotoxicity (overall mean CC50 = 72 µg/ml or ~ 2.4 µM) (Fig 2; Table 1, Assay 1)" (Lines 311-313). This is now also clarified in Table 1, footnote 2, “The IC50 and CC50 values given in the text for Assay 1 are the respective overall means for the four proteins”.

Fig. 1. Test of MAP30 and Momordin, with and without a Tat-CPP, in the A549-NLRV assay. Shown is a cropped screen shot from one Southern Research activity report. As can be seen, the four proteins have remarkably similar percent inhibition and viability values.

-I recommend including Figure S5 in the manuscript to illustrate the potential points of SARS-CoV-2 inhibition.

Both this Reviewer and the other Reviewer find this observation interesting. We initially also considered including this figure in the main manuscript, but we would prefer to leave it in the Supplementary Information as it is a prediction rather than a conclusion, and something that others may want to investigate. That is why we have put it in the Supplementary Information. Note: in addition to the 3dRNA structure predictions described in the manuscript, we have now also performed a more focused prediction on Hairpin No. 2 with FARFAR2. The results are essentially the same, giving us further confidence in our prediction (not shown).

-Legend of Fig S3. Model of MAP30-SRL interaction. (D-E) Orthogonal views of the MAP30-SRL complex. Panel F should be mentioned in the legend.

This was merely a typographical error; we intended to write (D-F), not (D-E). We have now corrected this.

---

## [Decision Letter · Decision Letter 1]

28 Apr 2023

PONE-D-23-02011R1The ribosome-inactivating proteins MAP30 and Momordin inhibit SARS-CoV-2PLOS ONE

Dear Dr. Watts,

Thank you for submitting your manuscript to PLOS ONE. After careful consideration, we feel that it has merit but does not fully meet PLOS ONE’s publication criteria as it currently stands. Therefore, we invite you to submit a revised version of the manuscript that addresses the points raised during the review process.

Please include the following items when submitting your revised manuscript:A rebuttal letter that responds to each point raised by the academic editor and reviewer(s). You should upload this letter as a separate file labeled 'Response to Reviewers'.A marked-up copy of your manuscript that highlights changes made to the original version. You should upload this as a separate file labeled 'Revised Manuscript with Track Changes'.An unmarked version of your revised paper without tracked changes. You should upload this as a separate file labeled 'Manuscript'.If applicable, we recommend that you deposit your laboratory protocols in protocols.io to enhance the reproducibility of your results. Protocols.io assigns your protocol its own identifier (DOI) so that it can be cited independently in the future. For instructions see: https://journals.plos.org/plosone/s/submission-guidelines#loc-laboratory-protocols. Additionally, PLOS ONE offers an option for publishing peer-reviewed Lab Protocol articles, which describe protocols hosted on protocols.io. Read more information on sharing protocols at https://plos.org/protocols?utm_medium=editorial-email&utm_source=authorletters&utm_campaign=protocols.

We look forward to receiving your revised manuscript.

Kind regards,

Israel Silman

Academic Editor

PLOS ONE

Journal Requirements:

Reviewers' comments:

Reviewer's Responses to Questions

**Comments to the Author**

1. If the authors have adequately addressed your comments raised in a previous round of review and you feel that this manuscript is now acceptable for publication, you may indicate that here to bypass the “Comments to the Author” section, enter your conflict of interest statement in the “Confidential to Editor” section, and submit your "Accept" recommendation.

Reviewer #1: (No Response)

Reviewer #2: All comments have been addressed

2. Is the manuscript technically sound, and do the data support the conclusions?

Reviewer #1: Partly

Reviewer #2: Yes

3. Has the statistical analysis been performed appropriately and rigorously? 

Reviewer #1: Yes

Reviewer #2: Yes

4. Have the authors made all data underlying the findings in their manuscript fully available?

Reviewer #1: Yes

Reviewer #2: Yes

5. Is the manuscript presented in an intelligible fashion and written in standard English?

Reviewer #1: Yes

Reviewer #2: Yes

6. Review Comments to the Author

Reviewer #1: The authors have provided responses to my first concern about the use of A549 cells from a retrospective (historical) manner. They have also included now in the revised manuscript more molecular simulation experiments which were placed in the Supplementary section. The authors also provided explanations on concerns about data discrepancy both in the responses to my concern and to another reviewer's questions. Overall the paper is now improved in terms of its scientific basis and presentation.

I still recommend that the authors make the justification on the use of A549 clearer in the manuscript, so as to help the audience understand the seemingly contradictory results in A549 and Vero experiments.

Reviewer #2: (No Response)

7. PLOS authors have the option to publish the peer review history of their article (what does this mean?). If published, this will include your full peer review and any attached files.

Reviewer #1: No

Reviewer #2: No

---

## [Author Response · Author response to Decision Letter 1]

3 May 2023

Response to Reviewers:

Reviewer #1: The authors have provided responses to my first concern about the use of A549 cells from a retrospective (historical) manner. They have also included now in the revised manuscript more molecular simulation experiments which were placed in the Supplementary section. The authors also provided explanations on concerns about data discrepancy both in the responses to my concern and to another reviewer's questions. Overall, the paper is now improved in terms of its scientific basis and presentation.

I still recommend that the authors make the justification on the use of A549 clearer in the manuscript, so as to help the audience understand the seemingly contradictory results in A549 and Vero experiments.

The Reviewer has requested that we justify our choice of assay, and to explain our results more clearly.

The assays were performed with both A549 and Vero cells. Both cell lines were being used at Southern Research to assay potential SARS-CoV-2 inhibitors. The A549 NLRV assay was expected to provide information about replication inhibition. The Vero CPE assay was expected to provide information on overall viral inhibition, due to both extracellular (entry) and intracellular (replication) inhibition. 

The results obtained with the A549 and Vero cell assays are not contradictory. Our data show a ten-fold difference between the 50% viral-inhibitory and cell-toxicity concentrations with the A549 cells and no difference with the Vero cells. Why the Vero cells did not “work” in this situation is something that remains to be determined. Our gel-filtration, affinity chromatography, and analytical ultracentrifugation results suggest that viral entry inhibition by blocking of the ACE2 receptor was not a factor. That the toxicity concentration for the A549 and Vero cells was very similar suggests that MAP30 entered both cells equally, and that something involved in viral replication is different in the Vero cells.

Our personal communications regarding this matter with Dr. Paige Vinson, Director of High-Throughput Screening at Southern Research, and with Drs. Timothy Sheahan and Ralph Baric at the University of North Carolina (the Baric lab is expert in this field and had provided both the A549 cells expressing ACE-2 and the USA_WA1/2020 nanoluciferase reporter construct to Southern Research to assay potential SARS-CoV-2 replication inhibitors), provided no explanation for why the Vero assay did not work. It may just be that human lung cells are more appropriate than monkey kidney cells in this case. Other cells, such as osteoclasts, keratinocytes, retinal ganglia, or follicular cells, may also prove unsuitable for assaying SARS-CoV-2 replication inhibition. A549 cells are commonly used to model the alveolar Type II pulmonary epithelium. They have been used to study various conditions including cancer, allergies, asthma, and respiratory infections. The A549 cell line has been approved by the FDA for use in a variety of applications, including in vivo testing of drugs and other therapeutics for use in clinical trials. We therefore think that the A549 assay data are relevant. As with many potential therapeutics, it is the differential between on and off-target effects that is of interest. Clearly, RIPs are not yet ready to be used as therapeutics, but our results demonstrate viral inhibition in a relevant cell type. When the molecular details of what is going on in the Vero cells is known, viral inhibition by RIPs will be better understood.

To provide more information to readers on how the cells were chosen we have added the following text at line 331.

“Both the reporter virus and the cells had been provided to Southern Research by the laboratory of Ralph Baric (UNC) specifically to assay potential SARS-CoV-2 replication inhibitors. A549 cells are commonly used to model the alveolar Type II pulmonary epithelium. They have been used to study various conditions including cancer, allergies, asthma, and respiratory infections.”

To clarify the results from the Vero assays we have added the following text at line 568.

“These observations suggest that SARS-CoV-2 inhibition by RIPs may be cell line specific and that perhaps in Vero cells some factor involved in viral replication is different from that A549 cells.”

---

## [Editor Report · Decision Letter 2]

16 May 2023

The ribosome-inactivating proteins MAP30 and Momordin inhibit SARS-CoV-2

PONE-D-23-02011R2

Dear Dr. Watts,

We’re pleased to inform you that your manuscript has been judged scientifically suitable for publication and will be formally accepted for publication once it meets all outstanding technical requirements.

Kind regards,

Israel Silman

Academic Editor

PLOS ONE
---

## [Editor Report · Acceptance letter]

22 Jun 2023

PONE-D-23-02011R2 

The ribosome-inactivating proteins MAP30 and Momordin inhibit SARS-CoV-2 

Dear Dr. Watts:

I'm pleased to inform you that your manuscript has been deemed suitable for publication in PLOS ONE. Congratulations! Your manuscript is now with our production department. 

Kind regards, 

on behalf of

Prof. Israel Silman 

Academic Editor

PLOS ONE